# A Sacred Place, an Enchanted Space: Crisis and Spiritual Elasticity in the Mountain of the Moon

**Clara Saraiva**

Institute for the Social Sciences, University of Lisbon, 1649-004 Lisboa, Portugal; maria-saraiva@campus.ul.pt

**Abstract:** This article explores the notion of the "magic of a place" and the way a space attracts groups and individuals who follow various forms of vernacular or lived religion and spirituality. The space is Sintra, an "enchanted" mountain facing the westernmost point of Europe, the Roca Cape. Classified by UNESCO as Cultural Landscape, Sintra is a unique place, a "sensuous sacred geography"; its sacredness comes from its natural setting, combined with historical layers of religious use and the way these are nowadays interpreted by individuals who live spirituality as "sensational forms" (configurations of imaginations and sensations in a context of religious and spiritual traditions). Thought of as an encapsulated magical place where innumerous groups perform their ceremonies, meditations, and spiritual retreats, Sintra is a scenario where Tweed's discussion on the sacredness of a place is highly suitable and transreligiosity and spiritual elasticity are the norm. Furthermore, through the ethnographic data presented, we will see how, within this "spiritual elasticity" directly relating to the astonishing nature of the Sintra mountain, individuals find relief for their personal crises or their collective eco-anxiety.

**Keywords:** Sintra; sacred places; lived religion; spiritual elasticity; crisis





*"Those who made us search for the ones who made them"*
Film *Artificial Intelligence,* robot talking to another robot.

## 1. Introduction: A Sacred Space

In this article, I will use the notion of the "magic of a place" and the way this space attracts groups and individuals who follow various forms of lived religion and spirituality. The space is Sintra, near Lisbon, known as the summer residence of Portuguese kings, which allies both monumental buildings (palaces, castles, chapels) with a lavish landscape. It was classified by UNESCO in 1995 in the category of Cultural Landscape. Beyond its heritage splendor and being a highly touristic spot, Sintra has been over the centuries a scenario for religious and spiritual ceremonies and rituals. Placed in front of the westernmost point in Europe, the Roca Cape, facing the Atlantic ocean, Sintra reveals itself to be an enchanted space. From the megalithic monuments to present-day shamanic walks, from Catholics to neo-druids, neo-shamans, Afro-Brazilians, followers of yoga, theosophy, and Eubiose, its special energy is used by everyone, offering a scenario of spiritual elasticity and transreligiosity (Panagiotopoulos and Roussou 2022), where vernacular, lived religion, and spirituality (Ammerman 2010, 2013, 2014) converge. Sintra's magic heals, comforts, assembles people, unites individuals with mother Gaia, and gives answers to personal and societal crises.

Following the "spatial turn", Kong (1990, 2001) underlined the importance of the intersection of sacred and secular forces in the making of a place and proposed further research on "sensuous sacred geographies" (amongst other themes) (Knott 2010, p. 30) and the "poetics of space" related to experience, aesthetics, the senses, and the sacred[1].

In this text, I will show how Sintra is a unique space that is both "generated and generative" (Tweed 2011, p. 117), where modernity, religion, and spirituality are entangled

(Bartolini et al. 2017) and how it also fits the qualification of a "sensuous sacred geography" (as proposed by Kong 1990, 2001 and discussed in Knott 2010) that suits the needs of the existing Portuguese religious pluralism and the way individuals (and groups) embrace a certain type of religious elasticity that relates directly to the search for well-being in connection to nature.

Knott et al. (2013) proposed the notion of the *secular sacred* to show how we must escape from a previous tendency to associate the sacred with the religious and therefore reinforce the opposition between secular and sacred. On the contrary, Sintra, as a UNESCO classified site, involves several secular aspects but has, simultaneously, its enchantment reinforced by the heritagization process (Testa 2020; Saraiva 2020). Redefining boundaries between the sacred and the secular enables marginalized groups to assert their existence and practices in public spaces especially amidst the religious diversity that is nowadays present in Portugal[2]. The flaws of the secular thesis (Casanova 2009) and the way the "spiritual turn" (Heelas and Woodhead 2005; Huss 2014; Fuller 2017) has to be reassessed on the basis of a redefinition of the borders between religion and spirituality that has been widely discussed (Ammerman 2010, 2013, 2014; Berger 2014; Davie 2000). Ammerman's (2013) research on being "spiritual but not religious" basically proposed going beyond binary categories that have long been part of the study of religion. In tune with the work by Fedele and Knibbe (2013, 2020), they all admit that the dichotomy between religious and spiritual is a valid one but propose to study such a divide as "a cultural phenomenon to study rather than as an analytical tool" (Fedele and Knibbe 2020, p. 1). In this text I will discuss, in line with what Fedele and Knibbe defend, how notions of spirituality go beyond such dichotomies and how we need a practiced and oriented approach, using the notions of "lived religion" that Ammerman proposed in line with what Birgit Meyer (2006, 2011) called "sensational forms" (configurations of imaginations and bodily sensations in a context of religious and spiritual traditions).

As we will see, Sintra is a transreligious space, where forms of vernacular, spiritual, and lived religion exist. Additionally, I will defend, following Panagiotopoulos and Roussou (2022), that religion and spirituality should not be perceived as antithetical; on the contrary, they must be approached "as two concepts with liquid boundaries, leaking transreligious fluid(ity) in-between their sacred spaces, and staining the rigidness of their religious boundaries with spiritual creativity" (Panagiotopoulos and Roussou 2022, p. 622).

Thinking that, in general, individuals are attracted to "sacred places" and to spirituality in search for solutions to all sorts of personal or social crises, I will use the notion of solastalgia. The word was coined by Glenn Albrecht (2005) and described as "the homesickness you have when you are still at home" and your home environment is changing in ways you find distressing. In many cases, this happens in reference to global climate change and the feeling that nature is changing and there is not much one can do about it. Individuals therefore seek refuge in a closer approach to nature, which becomes a spiritual healing vehicle also for personal crisis (Adams and Ghanem 2023; Annist et al. 2023; Whitlock 2023).

This paper is based on archival research, historical documents, and fieldwork from Sintra carried out from 2017 to the present date that comprised participant observation in walks and tours with the various religious/spiritual groups and in depth interviews with their leaders, guides, and followers. The voices of the individuals I worked with come out in the excerpts from the interviews conducted, showing how Ammerman's advocacy for a "practice approach" (Ammerman 2013, 2014), concerned with the lived religion and feelings of the social actors, is capable of unveiling how the construction of the sacredness of a place develops.

Amidst the variety of religious and spiritual groups present in Sintra with whom I have worked, for this paper, I chose the material relating to druids, shamans, yoga retreats, and Eubiose[3]. This sample will show how Sintra is indeed a transreligious space of "lived religion", how its sacredness was built in layers over the centuries, and how it is a place that I propose to call "transpiritual". Although, following entirely Roussou and

Panagiotopoulos' proposals, the term "transpiritual" seems even more suitable in the case at hand due to the types of religiosities that are explored in this text and the way in which individuals talk about it, as we will see.

I will start by explaining the various historical layers of Sintra's sacredness, giving some examples of places used over the centuries by various religious traditions (Section 2). I will then go on to the main focus of this article (Sections 3–5), which discusses my ethnographic data on Sintra's spiritual life, showing how this spiritual side is composed of intersected transreligiosities (as discussed by Panagiotopoulos and Roussou 2022), based on a strong relation with Sintra's nature, which helps people overcome crises and solastalgia. I will conclude (Section 6) by showing how such spiritualities can only be approached through a "lived religion" perspective.

## 2. The Spirit of a "Religious" Space

The enchantment of Sintra was built over centuries of use during which time the secular and the sacred have been in dialogue. In the 14th century, Sintra had already become (following the trend of the previous occupants, the Moors) the summer residence of kings and aristocrats. In front of the Roca Cape, with the fresh winds coming from the ocean and away from the capital but not too far apart (about 35 km from the Lisbon center), it became famous as a safe area to escape the plagues (that from time to time scourged the city) and as a hunting ground. As the aristocracy followed the trends of the court, the number of palaces and manor houses augmented in number and grandiosity from the 16th century onward, attaining a peak in the 18th and 19th century. In the Romantic period, Sintra became part of the Grand Tour, the voyage undertaken by young male aristocrats as a sort of rite of passage marking their coming of age and the completion of their education (Saraiva and De Luca 2021; Cardeira da Silva and Saraiva 2022).

While Heelas and Woodhead (2005) defend their subjectivation thesis concerning present-day spiritual forms, Charles Taylor expands on expressivism, underlining how Romanticism and its "philosophy of nature as a source. . .a force, an élan vital running through the world, which emerges in our own inner impulses" (Taylor 1989, p. 373; Sointu and Woodhead 2008, pp. 263–64) was important in the development of a certain line of thought. Sointu and Woodhead (2008) defend that Taylor's critique of contemporary forms of expressive selfhood seems to be shaped by a lingering attachment to the masculinist hero of Romanticism.

In the case of Sintra, the hero of Romanticism had a face: Ferdinand II, a prince from the Saxe–Goburg–Gotha dynasty, who came to Portugal to marry Queen Mary. A pure Romantic, he fell in love with Sintra, bought the old Hieronymite convent on top of the hill, rebuilt it (as it had been partially destroyed by the 1755 earthquake and abandoned with the extinction of the religious orders in 1834), and transformed it into a fairy-tale palace. With his love for nature, he also had an enormous park created in the adjacent hill and had the mountain reforested. Ferdinand was a romantic, a lover of the arts, and a Catholic; the alabaster altar in the convent chapel was carefully preserved. Portugal was then (as nowadays) a Catholic country and the religious elements in the Sintra area attest to such tradition. Other than the chapels present in every built palace or manor house, the monuments that testify to such religious scenario are abundant, as are the several convents that housed various religious orders. In addition to the referred Hieronymite monastery on top of the hill, built in 1511, replacing a hermitage dedicated to Our Lady of Penha where Ferdinand II was to build his grandiose palace, already in the 12th century the Romanesque chapel of Saint Saturnine had been erected on top of the Peninha hill and overlooking the Roca Cape. The Penha Longa convent was constructed in 1355 and also donated to the Hieronymite order in 1390.

The Capuchos convent of the Holy Cross, built in the 16th century, is the most striking case of asceticism founded in Christian principles. The convent is also known as the Cork Convent, due to the profuse use of cork in its interiors. Its setting amidst oaks and cork-oak trees, boxwoods, and chestnut and hazelnut trees is one of the few areas in the Sintra hills

where the original vegetation has been safeguarded. The construction is very simple and uses the immense natural boulders as walls. The rusticity and austerity of the construction dialogued with the life of suffering and atonement of the monks, who followed the ideals of the Order of St. Francis of Assisi and the Capuchin rules (the search for spiritual perfection, alienation of the world, renunciation of all pleasures associated with earthly life, extreme poverty, allied with the mortification of the flesh, and the mystical values of hermitic life (Saraiva and De Luca 2021)). Built in respect for the harmony between human construction and the pre-existing natural elements, it is often described as a "divine construction" that helped the friars find their path to ecstasy (Muchagato 2013). As happened with other monasteries, the convent was abandoned in 1834, with the extinction of the religious orders determined by the liberal regime. As part of the UNESCO world heritage site since 1995, it recently went through a major rehabilitation and is now open to the public.

Although such monuments all attest to the Catholic devotion in Portugal, the fact is that, as overall in the Catholic world, many of these spaces had previously been local sites where ancient religions and spiritualties performed their rituals. As elsewhere in Southern Europe (Blanes and Mapril 2013; Saraiva 2013), such "pagan" traditions were later on absorbed and transformed into Christian uses and sites. The chapel of Santa Eufémia, a 19th century shrine situated on another hill top, was a place of use of holy waters and was already inhabited in 4000 AD, where the Celts[4] built a temple dedicated to the moon; the Sintra hills are known as the Mountain of the Moon. In 1147, a hermitage praising such medicinal properties of the local waters was erected. It is reputed to be the origin of the cult space (Cardim Ribeiro 1998b; Saraiva and De Luca 2021).

Another chapel was built in the 16th century in Janas, replacing a temple dedicated to Diana, the Roman guardian of wild animals and the hunt. The 16th century chapel has a unique rounded shape and was dedicated to São Mamede, the Catholic protector of cattle (Rodil 2018). The popular *romaria*[5] continues to take place every year and the blessing of domestic animals, such as as dogs and cats, has been added to the blessing of cattle. In Penedo, the feasts in honor of the Holy Spirit, installed in Portugal by the Holy Queen D. Leonor, known for her charity deeds, and sanctified in the 16th century, also have pagan origins and used to include the slaughter of the bull as a sacrifice for the divinities, with a later Christian touch added to prepare meals for the poor. Other *romarias* abound in a country where the religious landscape has been dominated by Catholicism for centuries.

Going back still further in time, the most famous pre-historic monuments (amongst others) are the *Adrenunes dolmen*, hidden in the woodlands amidst gigantic granite rocks and facing the Atlantic and the Roca Cape; the *Tholos dos Monges*, a burial site from 4500 years ago; the dolmen of Agualva and the dolmen of Bellas. From Roman times, the Alto da Vigia, in the cliff over the ocean, is considered to have been a Roman temple dedicated to the eternal sun, the moon, and the ocean[6]; it was also used during Islamic times (Saraiva and De Luca 2021).

The reputation of the "Mountain of the Moon" as a special energetic place and an idea of the supernatural relating to personal connections or to legends and fantastic stories survived throughout the ages and found a fertile ground to grow during the Romantic era. Famous writers mention the energy felt upon approaching Sintra, either from the Atlantic side or coming out of the capital into the sacred woods. As Robert Southey put it when describing the "magical mountain": "(...) had I been born in Sintra, I think there would be nothing that would tempt me to abandon its delicious shadows and go through the terrible dryness that separates them from the world". William Beckford also wrote about Sintra as a "magical place, (...) I believe myself in the garden of the Hesperides, to expect the dragon under every tree. (...)."[7] (Beckford 2005) and Byron glorified it as "The Garden of Eden". The imagery of this encapsulated and magical site that one enters thus incorporates Eliade's notion of the space as a site of the sacred and, as we shall see, the awesomely fearful and the enthrallingly captivating aspects of "The Holy" (Eliade 1959) and of Otto's "mysterium tremendum" (Rudolf Otto's notions), all foundational aspects of the sacred (Saraiva and De Luca 2021).

If the use of the Sintra mountain for rituals and religious ceremonies dates back to Neolithic times and its mystical magical aura is present in the megalithic and Celtic testimonials, such appeal has been revitalized in the last thirty years. Several new religious movements[8] use the *serra* for their rituals and ceremonies, from neo-druids, neo-shamans, neo-pagans, Buddhists, and yoga followers, to Afro-Brazilian religions or masonic movements, most of them falling into the Fuller (2017) classification of secular spirituality (Saraiva and De Luca 2021). Such groups have found their place as minority religions in a landscape characterized by an increasingly religious pluralism and legitimization of new forms of spirituality in a country where the Catholic hegemony was the rule (Di Placido and Palmisano, this volume; Roussou 2021; Saraiva 2016, 2020).

Now, Sintra has a double life: crowded with tourists attracted to the splendorous UNESCO site during the day and groups of individuals who embrace various spiritualties by roaming the mountain mostly at night. Sintra is a highly touristic site, where tensions between locals, religious groups, and the management of the UNESCO site abound (Saraiva and De Luca 2021; Cardeira da Silva and Saraiva 2022) and conflicts over the various heritage regimes (Bendix et al. 2012; De Cesari 2013; Geismar 2015; Hafstein 2012; Hafstein and Skrydstrup 2020) are continuously present.

As Thomas Tweed (2011, p. 117) stresses, spaces are differentiated, interrelated, and have a kinetic nature as they involve movement and processes that change over time. Tweed defends that "space" is a more abstract notion, while "place" is a more defined notion linked directly to lived experiences. This is the case of Sintra. If religious spaces are particular spaces, Sintra is a (religious) *space* that became a very special *place*, defined by the multiple experiences of people throughout the centuries. Sintra has its history, with religious spaces that are interrelated; Christianity is one of them, along with many others.

As the above examples show, Sintra is a sacred place generated by layers of spiritual uses and religious occupations; simultaneously, it is also a generative place. Sintra's space *does things* (Tweed 2011, p. 229), as we will see. It acts as a "powerful magnet"[9], attracting a multitude of spiritual groups and individuals who look for inner-knowledge and reconnection with nature to alleviate personal or collective (planetarian) crises. Here, the concept of "nature" is taken at face value, in the broadest sense, as the physical world or universe. This natural world is not a passive reality but a natural environment with its own life. It can "do things" to people. This is the case of Sintra, where its natural environment can help and heal people, as we shall see[10].

## 3. Secular Bodies, Not So Secular Practices

### 3.1. Sacred Paths

In the summer of 2022, I went to Sintra to meet a colleague who was attending a yoga retreat at one of the manor houses placed in the midst of the mountain. To reach it, one has to follow a "private road" with a gate at the entrance and another one at the exit, with a very clear sign stating "Private property. No Entry" on one end and a gate, carefully locked with a large metal padlock, at the other end. If one passes the first gate, one penetrates a magic road full of old cedar trees and some redwoods that hardly let the sun come through. One enters a magical and wonderful route of connection with nature that also resembles the vaguely kitsch photos in a religious calendar or book; a few rays of sun come through the woods and attest to the glory of the Creator.

When I finally reached the manor, there was a group of 15 people at the yoga retreat. The retreat was organized by a young couple who do this for a living. The "clients" were foreigners in their great majority. Yoga sessions took place in a room with large glass windows from which one could glimpse the Pena palace. I was also there for an Indian concert, where a young Spanish girl sang several traditional Ayurvedic songs. Everyone sat on the floor; some seemed to enter a semi-trance posture as they swung their bodies to the music. In the last song, everyone stood up and sang and danced together. There was definitely a sense of communion in the air and a feeling of shared sentiments. Although these people were there together for six days, I was only there for a couple of hours.

Such retreats happen often in the Sintra area. Under the heading "Take a break and experience a divine body rest and unique peace of mind. Give yourself a gift of love and relax; you deserve it." There are even very expensive mindfulness retreats[11], appealing to a getaway meditating retreat to redirect individuals back to a relationship with Mother nature. Another advertisement states "Retreat in Sintra. Portugal Silent Meditation Retreat", "During these 5 days, you will join a three-day dive into the Silence of the Heart. You will gradually explore the foundations of meditation practice, discovering the joy of stillness and the intimacy of the heart", "the instructor has been connecting with various spiritual traditions throughout his life", "The retreat will take place at the foothills of the sacred Sintra mountain, in Lisbon, Portugal".

### 3.2. Running with a Stick

That night was one of the many nights when I (and 12 other individuals) followed the shaman through the narrow trials of the Sintra forest. During many parts of the walk, we could barely see the trail since the woods and bushes had grown and hidden the way, but Armando[12] knew exactly where to lead us. We were walking silently and in total darkness, with no pocket lights or mobile phones, hand in hand with the others so that no one would become lost or fall without the others noticing. The sky above us was dark (no moon that night), but, as our eyes became used to the lack of light, the silhouette of the tall trees became evident, millions of dark arms and leaves above our heads. We made several stops to meditate, touch the soil, and feel Mother Earth. Towards the end, as we had to come down a larger trail, Armando told us to run down energetically, lifting our knees high, holding our walking sticks (those who had one), and arm in arm with our neighbor to feel the energy of the place. I thought to myself I was definitely going to fall and end up with bruised legs on the dirt floor running down the hill quite fast, holding a tall wooden stick, arm in arm with Armando.

These walks are shamanic walks that last about four hours, starting at around 11 p.m. The trails vary, but they are always intended to be a total immersion in the magic of the Sintra forest, in silence, to allow individuals to feel in contact with nature and, their inner-self, but also to feel communion and sharing "something beyond us" (as one of the individuals told me) with others.

### 3.3. Discussing Sacred Paths

In both cases, this spirituality that we see in Sintra nowadays is out of the public realm (Fedele and Knibbe 2020, p. 3), hidden behind the walls of old palaces where yoga retreats take place or in silent night walks. As we saw above, the religion that is public is the hegemonic Catholic religion. The temples that are visited by the tourists are Catholic ones and the processions that are entitled to escort by the police are Catholic ones. In this sense, sometimes the spiritual overlaps with religion, other times it does not. Sometimes individuals do not mind if they are called "religious" or "spiritual", other times they react violently. For many, as Fedele and Knibbe argue (Fedele and Knibbe 2020, p. 2), defining themselves as "spiritual" is a clear reaction against a very Catholic upbringing, a fight against institutionalized religion that they feel used to suffocate them. It also makes them feel closer to what academics often call "a secular approach", but which, for many individuals, is just a way to feel free to choose their own ways of connecting to the supernatural. Having all this in mind, spirituality is here looked upon from a perspective of lived religion (Fedele and Knibbe 2020, p. 2), which offers solutions to life crisis situations, e.g., the usual ones concerning love, work, financial problems, and health (Saraiva 2011), but also the connection with Mother Earth. For many of the individuals I interviewed and walked with, it is the loss of connection with nature from which such problems arise. Reconnecting with Mother Nature in such a magical place helps to recuperate one's balance.

Nancy Ammerman argues that spirituality is more contentious and less easy to categorize than religion and that, in the US, it is also a "discursive category defined by various ways of encountering 'something beyond'" (Ammerman 2010, p. 157). Although the

Portuguese context is quite different from the US one, such statements are still valid and I fully agree that religious pluralism is "endemic in the human condition" (Ammerman 2010, p. 166) and that it is the normal (and not new) state of human affairs. Different to the US, Portugal does not have a history of religious freedom. It has been, over the times, officially and non-officially a country of Catholic hegemony. Catholicism was the religion of the Portuguese kings and queens from the foundation of the kingdom (1147) until the triumph of the republic in 1910. Even the turmoil of the late 19th century liberalism did not erase that hegemony and several concordats were celebrated with the Vatican (Vilaça 2006; Vilaça et al. 2016; Vilaça and Oliveira 2019). The Estado Novo dictatorship, implemented from 1933 onward, gave the Catholic church a new centrality and reinforced the "Christian Reconquest" (Dix 2010, p. 12) in the 20th century. Under the Salazar rule (1933–1974), the Catholic church managed to keep its status of privilege and religious minorities were not welcomed by the strong nationalist regime and were even persecuted (Saraiva and De Luca 2021, p. 161).

This scenario changed with the 1974 revolution, which marked the end of the dictatorship and Portuguese imperial domain and the subsequent new constitution (1976), which consecrated the principles of religious freedom. Such ideals were further reinforced with the 2001 law of religious freedom. Mostly from the late 1970s onward, with the arrival of populations from the former Portuguese colonies, and from the 1980s on, with the entrance of Portugal into the European community, with the input of people coming from all different continents, societies, and religious affiliations, the country definitely became the locus of a multicultural and multiethnic society (Saraiva 2013). What we have nowadays is a diverse religiouscape, where many groups exist side-by-side, especially in the two largest metropolitan areas of Lisbon and Oporto. We can find Brazilian charismatic Catholicism, Jewish congregations (Pignatelli 2020), Punjabi Sikh or Hindu temples, Islamic groups (Mapril et al. 2019), evangelical, neo-Pentecostal (Mafra 2002), and African churches (Blanes 2009; Sarró 2009; Sarró and Blanes 2009), Afro-Brazilian religions (Pordeus 2009; Saraiva 2008, 2013, 2016, 2020), orthodox (Vilaça et al. 2016), Buddhists (Vilaça and Oliveira 2019), as well as neo-pagan, neo-shaman, and neo-druid groups (Fedele 2013; Roussou 2015, 2017), amongst others.

## 4. The Sea, the Forest, and the Magic of Sintra

Both those responsible for the yoga retreats (e.g., Paula and Zé) and the individuals who frequent such retreats believe that Sintra is indeed a space with a special energy. The couple in charge of such happenings defend that Sintra has an ancestry of spiritual tradition connected with the historical layers of the populations that inhabited the area from Neolithic times, through the Celts, Romans, Moors, etc. The choice of the place for their retreat site was Sintra due to this magic related to the natural forces of the ocean and the forest. Many of the meditation sessions take place at sunset at the beach or on a cliff facing the Atlantic.

Other than the basic practices related to a subjective-life spirituality and the search for the inner-self (Heelas and Woodhead 2005), such as relaxation, massage sessions, meditation sessions, and yoga practices, they include in the program sessions directed at the search of the inner-self, such as the "systemic family constellations", a psychotherapy based on the three laws of love: right of belonging, hierarchical order, and balance between giving and receiving. One of the highlights of the retreats are the walks for forest therapy and forest bathing: "A forest is the climax of the evolution of an ecosystem. The Sintra forest is a natural setting in which the tranquility it provides allows us to absorb more nutrients, and gain our force!" as Paula phrases.

The majority of the clients are not Portuguese[13]. In the large room, with a large table, sofas, and reading materials dedicated to daily conviviality amongst the guests, there is a world map on the wall with pins showing the various origins of everyone who has come here for the past fifteen years. The pins indicate guests form Ukraine, Poland, Turkey, Iraq, Iran, Saudi Arabia, and Egypt and of course, many from Canada, the US, and Central and

Northern Europe. How do so many people coming from the five continents end up in this final stretch of Southwest Europe? Paula explains that people come here in search of connection: "Most of them are prisoners of their own stories and loneliness. They live in cities where there is a huge disconnection with nature and with ourselves. Often individuals are also concerned with the state of the planet. The fact that they are not well with themselves has a reflection on their relation with the rest of the living world and the planet".

Although conscious of the mercantilization of well-being, Paula and Zé believe that this recent industry of well-being is better (for people and for the planet) than other forms of tourism industry, where people just consume the places without reflecting upon themselves, their lives, or the relation with Mother Earth. The voice of the individuals who attend such retreats in Sintra corroborate such ideas:

> "All we go through here makes one feel in tune with nature, and this is indeed a special, magic place. Sintra makes us reunite again with the sacredness of nature. We come to understand this here, and are therefore able to change it or improve that relation. Meditation is in fact a state of consciousness, and it helps one in developing the commitment to relate to Mother nature".

The term solastalgia was coined to draw attention to negative psychological impacts caused by the destruction of one's home environment (Albrecht 2005) and feelings of loss, stress, and dislocation associated with ecological disturbances[14]. It is used nowadays in contexts that recognize feelings of loss and negative mental health associated with the impacts of climate and environmental change (Adams and Ghanem 2023, p. 14). According to these ideas, humans can only exist in relation to places around them and emotional bonds to such places are part of the human condition (Tuan 1975 in Adams and Ghanem 2023, p. 3.) The feeling that one is powerless facing natural disasters that occur also due to human fault triggers sadness, anxiety, distress, anger, grief, stress, and other emotional states and psychological distress related to eco-anxiety (Galway et al. 2019, p. 300; Shrisvastava 2023, p. 151). Individuals have the conscience that, in order to prevent the consequences of climate change, new choices and plans for the future have to be made (Annist et al. 2023, p. 302), but they feel quite powerless in the face of world political decisions. One way to overcome such anxieties and to balance or offset depressive feelings that affect many youngsters nowadays is to cultivate moments of joy, awe, or delight (Whitlock 2023, p. 298). Reflecting upon such issues and becoming more aware of the dangers already makes one feel better: "The awareness is sad, but at the same time revigorating. . .we are no longer amorphous, we understand what is going on. That is the first thing to do. . .it is impossible not to feel it when you walk through the woods of Sintra. . ." explained one of the participants in the yoga retreat.

Sintra's powerful nature is a solace to those anxieties. The atmospheres of the yoga retreats or the night mystical walks create elicit feelings of well-being and peace of mind in tune with Mother Earth[15]. Such conceptions of spirituality are also political statements. As Fedele and Knibbe put it: "The injunction to 'listen to yourself', pay attention to how you feel may sound narcissist, but is its subversive when it is addressed to a self whose perspectives and experiences are usually dismissed" (Fedele and Knibbe 2020, p. 13). The holistic spirituality practiced in Sintra falls into the category of "forms of practice involving the body, which have become increasingly visible since the 1980s, and that have as their goal the attainment of wholeness and wellbeing of 'body, mind and spirit" (Sointu and Woodhead 2008, p. 259)[16]. Individuals feel better and alleviate their tensions or distress in these walks.

The guide who leads the forest immersion was carefully chosen in order to have someone abiding to the atmosphere of the retreats. She is a biologist who values what she calls the "intelligence of the plants". She defends that her mission is to re-connect people with nature and that the port of entrance for this re-connection are the trees, the lichen, and the water: "This is my spiritual part, that I try to transmit in the walks: we

are all connected". Cristina complains about what she calls "plant blindness", the fact that individuals nowadays now nothing of the plants that surround them; by the end of the walks "often individuals come directly to me to thank me, and say how much they have learned about the plants and nature, and how that makes them feel so good". She explains that she always refers to "forest bathing" and how that is an excellent way to overcome anxiety:

> "the results of two hours in contact with nature are overwhelming: the blood pressure comes down, it reinforces our immunity system, and the benefits last for months. I do this as I encourage them to stay quit and fell nature" (…)"I try to create a consciousness in the individuals, and explain how trees, for instance, furnish oxygen and clean the carbon dioxide, but how they also release other beneficial substances for us, humans, and for the planet in general".

As others, she emphasizes the uniqueness of the magic of Sintra:

> "I recall vividly a night long walk. As we arrived at Peninha at sun dawn, there was still a bit of moon, and a magic purple haze over the sea…we waited until the sun came up, and it was sheer magic. This is what, for me, is spirituality. It was very introspective and personal, but also shared; we even shared aloud our life intentions. After a whole night walking basically in silence, that was something" (…) "I do feel that Sintra has a special energy. (..) it works like a magnet (…) very strong".

Blaming the evil in the world for the lack of connectiveness (with ourselves, with the others, and with the natural world that surrounds us), she thinks of herself as "a spiritual person, based on the relationship with nature, which is part of my life (…) the natural world has always exercised a fascination over me; and that is my mission, to connect other individuals to that fascination".

The night walks, organized by Armando (who follows a shamanic route), elicit the same type of feelings, as participants explain:

> "In the beginning I felt fear. The whole idea of walking in the middle of the forest, with no lights, with people I did not know…but, as we went along, I went from fear to trust…feeling I was letting my body go, and I felt (…) a confidence that translated into serenity (…) a feeling of spirituality that was not an escape, but another way to look at and deal with anxieties and fears".

Night walks engage in a profound relation with nature and with what individuals perceive as the *magic of Sintra*: "Night tours are a particular example of the blurring confines of categories of secular and spiritual when they are projected on the environment" (Saraiva and De Luca 2021, p. 172). Certainly, they play with the right dose of magic and mystery, both essential in underlining the power of religion, spirituality (Ammerman 2010, p. 158), and the sacred, as one of the night walker's commented:

> "I am absolutely sure all this had to do with Sintra. Of course one hears a lot about the magic of Sintra and that influences us. But it was Sintra's magic indeed. (…) An involvement by nature and *by the force of nature* that is Sintra's nature".

Other night walks organized by the Order of the Bards, Ovates, and Druids (OBOD), elicit the same ideas. Although individuals search for their own inner-directed paths, following a path in a group and sharing emotions and sensational forms (Meyer 2006, 2011) is a way to arrive at inner peace.

On one such night walk (February 2023), the guide explained that the walk would be a visit to parts linked to mysticism and the imaginary and that it would include paths in the mountains and village. As we continued walking through the forest, she went on to talk about several paths that incorporated the Camino de Santiago, about the secret tunnels that connect Cascais and Boca do Inferno to Sintra, and about the imagery of the mermaids/nereids (sea nymphs that help fishermen and sailors): "Sailors said they heard

the Nereids singing and asked them for help if needed. They came through the tunnels and transformed themselves, in the mountains, into forest nymphs". She also explained how followers of ufology also believe that there is a people who live in the holes of the mountain (a people of giants) and the holes in the earth relate to the idea that there is an intra-uterine world, an underground Sintra accessed through tunnels. She clarified that Sintra is also a center of power through which ley lines pass[17], imaginary lines that unite this space of Sintra (the path we were following, which is next to the initiatory well of Quinta da Regaleira) to other centers of energy in Portugal (and elsewhere).

The discourse continues regarding the mysticism in the mountains, the Neolithic and prehistorical rituals, Satanism and witchcraft, voodoo and African knowledge of the mountain, legends of enchanted Moorish princesses and of Our Lady and of lost souls and reconducting to mythologies related to the Portuguese triumph of Christianity over the Moors (Cardeira da Silva and Saraiva 2022). The menu is immense and wanders through all possible imaginable stories.

Even if the past is contested, conflictual, and much constituted (Meskell 2012, p. 1), sometimes the past is also a conflictual arena that does appear in the discourses and seems to be out of the consciousness of the interviewed. Issues of collective memory and religious forms and practices (Isnart 2020; Isnart and Testa 2020, p. 5; Testa 2020, p. 20) come together and there is a symbolic capital coming from the past that is used in diverse ways, connecting the realms of politics and religion (Isnart and Testa 2020, p. 6; Saraiva and De Luca 2021, p. 153). Individuals following the new spiritualities (druidism or neo-paganism) go on night spiritual walks or conduct yoga retreats (just to name a few) to contest the politics and management of the UNESCO site and complain about its interference with their larger civic and religious freedom. Still, and apparently in a contradictory way, they also talk about the special Portuguese spirituality of the Fifth Empire, a legendary story that entangles visions, Catholic saints, and Portuguese myths of the rediscovery of Portuguese nationalism. In fact, Sintra elicits discourses of legitimacy that appeal to real or imagined historical and religious pasts, which feed the production of various narratives that are often conflicting (Saraiva and De Luca 2021, p. 167).

If in Sintra an aesthetic Orientalism (Said 1978) directed at tourist consumption is displayed, even new agers and tour guides outside of the main touristic frame invoke a certain relation to Sintra's magical heritage to an exceptional and grandiose Portuguese past (Cardeira da Silva and Saraiva 2022, p. 170; Bastos 2018), even without consciously acknowledging it. The essence of the *magic* of Sintra appears to elicit a consensus based on sensational forms (Meyer 2006) that are sustained and shared and modes of invoking and organizing access to the transcendental, which offer structures of repetition to create and sustain links between believers.

The site of the OBOD explains that "the Druidic Tradition is ancient and represents one of the sources of inspiration of the Western Spiritual Tradition. But while it is ancient, it is as relevant and alive today as it ever was. Druidism (...) is now experiencing a Renaissance which is for some a spiritual path, for others a religion and for still others a cultural activity"[18]. The leader of the group believes that druidism has become a vital and dynamic nature-based spirituality related to "the love of land, sea and sky—the love of the Earth, our home." In the interviews conducted, he states how he believes Sintra has indeed a special energy and that it is the pre-historical aspects, followed by other occupancies and religious uses (Romans, Celts, and Lusitanians) that give a "unique energy" to the place. Places such as Anta de Adrenunes, Peninha, Tholos do Monge, and Bosque dos Druidas became special places to feel the "energy of the place" and to "apprehend transcendence", since the rocks—abundant elements in these places—keep the memory of the ancestors who were there. These spaces gain relevance for being a link between past and present and they create connections with the people who inhabited the region, in addition to providing support for ritualistic practice.

To exemplify, he considers Santa Eufémia, located in the Parish of São Pedro de Penaferrim, one of the "strong points" in the mountain. He explains that Santa Eufémia,

similar to other places in the region, went through a process of Christianization. He also states that, to feel this strong energy, it is necessary for a person to have serenity, which is often unfeasible due to the excessive number of tourists. Most of the OBOD celebrations are held in the druidic temple, built in the secluded Quinta dos Lobos and thus offering an environment of tranquility. He explains how druidism values meditation, which is "fundamental for us to feel what we call magic, the spirit of the place. Everyone feels or doesn't feel, in their own way. It's a very intimate thing. Our society devalues intimacy. Everything is for display, everything is for sale. (...). This space, in terms of spiritual work, is one's sacred grove, in which the person visualizes the sacred space within oneself, where one meditates, prays, one's intimate space".

He finishes his reflection on the "spiritual revolution" (Heelas and Woodhead 2005) brought by the druidism revival in Portugal, asserting the uniqueness of Sintra: "Sintra is a very, very special place. I have travelled a lot in Europe and I have seen amazing places (...) But for me, is unique. It's a mountain that isn't very big, it is not very high, but it has something that transcends language. (...) Sintra's energy is different from other places".

## 5. Theosophy and Eubiose: The Secret Tunnels of Sintra

For the religious groups of our ethnographic research, the *Serra* of Sintra and its tangible heritage sites are primarily spiritual sites. The UNESCO classification has simply confirmed this enchantment in secular terms. They maintain that it is the spiritual energy of Sintra which has made it a historical privileged place *before* and *beyond* the spatial boundaries drawn on UNESCO's maps.

In their discourses, the heritage component collapses into the spiritual and is reinforced by it. For this reason, while a few appreciate the forms of protection of the heritage sites implemented by the local administration and a semi-private enterprise, others complain how the economic interests that underlie patrimonialisation policies threaten and overpower their religious freedom (Saraiva and De Luca 2021).

One of the women initiated in OBOD defends that the Mount of the Moon (Monte da Lua) has "a very specific energy, very feminine, with great magnetism, which is why people come here, looking for this". She organizes day and night walks, with titles such as "Extraterrestrial Sintra" or "Haunted Sintra", making use of certain special places (such as the chapel of Saint Saturnino in the Peninha cliff and AdreNunes), stressing the link of the divine to Mother Nature.

Bound at once to the flow of historical events and to the timelessness of the spiritual plane, for many groups that roam the *serra*, Sintra and its architectures are all part of a cosmology that puts it in connection with broader spatiality. "The mystery of the spiritual discovery of oneself can lead to the revelation of the centre of Mother Earth, which, in the case of Portugal, corresponds to Sintra" explains Vasco, a studious and charismatic guide of a religious community founded in Portugal at the end of the 1970s, and that he defines as "parent of Christianity and close to freemasonry", with several publications on the specific energy of the place:

> "... what marks Sintra, what is unique about Sintra, is its spirituality; the energy centre, and really the climate here, the energies of Sintra are very special, the land is lavish [...] the air is charged with that life energy. And Sintra really stood out as one of the vaults, the spiritual centre of the world".

According to Vasco, what he refers to as the "sacred mountain" is one of the Vedic chakras of the planet and the axis-mundi of Western Europe: "the true Arcadia of the Gods, where the presence of the Kingdom of Agharta[19] is intense"[20] (Adrião 2017, pp. 327–29). In several of his books on the subject, he explains how Sintra is an ancient volcano and constitutes therefore an immense magnetic massif[21]. Following the doctrine of Henrique José de Sousa, the founder of Brazilian Eubiose, related to Blavatsky's theosophy, he also defends that Sintra is one of the seven places on earth that drains the energies coming from the core of the planet. Several spaces in Sintra are meant to be the exact spots for such

drainage. Such spaces coincide with the ones I referred to in the beginning of this text: Santa Eufémia; the chapel of Saint Saturnine facing the Roca Cape; the Pena palace/ancient Hieronymite monastery.

Doctrines such as Eubiose and theosophy, once again, relate to an international idea of relation with the universe as a whole. The Brazilian Eubiose site is fertile, with short videos explaining the origins of the movement and also images stating that Eubiose is "to live in perfect harmony with the laws of the universe, and to transform energy into conscience; it is self-knowledge; to reconnect the various dimensions of the sacred, the profane, the divine and the human, aiming at the elevation of human consciousness".

Stressing the uniqueness of Sintra in Portuguese soil as the exceptionality of Sintra in the formation of Portuguese spirituality, Adrião simultaneously appeals to a pan-enchantment of Vedic chakras and the relation of Sintra with those other sites whether in India, Tibet, or elsewhere.

## 6. Personal Crisis and Experiences of *Mysterium tremendum*

For all these users, Sintra perfectly embodies the notion of place as "the most fundamental form of embodied experience—the site of a powerful fusion of self, space and time" (Feld and Basso 1996, p. 9). We are facing what Kong defines when she discusses the concept of "sensuous sacred geographies" (Kong 2001; Knott 2010). The sui generis nature of the sacred (Eliade 1959) relates directly to the experiences, aesthetics, and senses. Indeed, Sintra is a "numinous space" that conveys experiences of transcendence. This can be expanded to think of Sintra as a multivocal and polysemic scenario that is both generated and generative (Tweed 2011, p. 121). Sintra is a space that became a special place (Saraiva and De Luca 2021, p. 153). It was constructed over time and is reinforced today by both visitors and religious users of the sacred mountain. With its special energy, Sintra is believed to be divine and falls into what many new agers would call a "centre of light" (Ivakhiv 2007). In this conception, the earth is akin to a living organism, with its own energy, consciousness, and intelligence, as well as a strong sacredness (Rocha 2017, p. 137). This is exactly what individuals refer to and they would certainly agree with Casanova's (2009) perspectives concerning the secular and the religious, stating that, in Sintra, nothing is secular, as even a possible secular view is endowed with the enchantment of the notion of heritage as a sacred being (M. Macdonald 2003; S. Macdonald 2013).

We conclude that everyone is correct when we reflect on the example of the spiritual use of Sintra: Ammerman when she states that religious pluralism is the norm and finding that the theory of secularization has long ago fallen apart; Casanova when he defends that there is no clear cognitive difference between science, philosophy, and theology (Casanova 2009, p. 1051) as the religious groups and individuals that praise Sintra blur the concept of modern secularism. This ties into the notion of secular spirituality, referring to the potential for all experiences to assume a spiritual quality not limited to any one religious or transcendent realm (Saraiva and De Luca 2021, p. 167). In Sintra, we find many forms of contemporary religiosity, embracing not only most of secularism's basic premises (Fuller 2017) but also Ammerman's idea that religion (or, should we say, spirituality) "does not stay neatly in a cosmological box" (Ammerman 2010, p. 161). Such concepts form a mix of secular spirituality (or spiritual secularism?) that relates to self-knowledge and self-growth, openness to wonder and awe, metaphysical explanations, and communal and ecological morality to face the disasters that humans have infringed upon Mother Earth. The individuals in the groups we have analyzed believe that there is indeed a spirituality, "something beyond us", most of the times impossible to explain but which is felt, a sense that the spirit has reasons of its own and that the key to re-connect is to re-organize their relationship with nature, in this specific case, with Sintra's nature. Heelas and Woodhead's (2005) descriptions of a "holistic milieu" that is both individual but also communitarian apply here. No matter how personal the paths may be, the search for a "pure" feeling of the sacred nature of Sintra implies a sense of community, as Rocha (2017, p. 9) defines following

Appadurai (1996, p. 8), "a community of sentiment, a group that begins to imagine and feel things together" and as one of the participants in one of the night walks stated:

> "At first I was afraid of what the others might think of me—tripping, not knowing what to do during the walk, also when we had to say a few words about ourselves, my insecurities arouse. . .—but as the hours went by one feels a sense of togetherness, we are all there for one common purpose, and when, towards the end, we embraced each other, it was as if I had always know those individuals---whom I had never set my eyes on five hours before".

In her book on religious experience in Brazil, Bettina Schmidt highlights the contribution of the classical text of William James *Varieties of Religious Experience,* where he stresses the importance of the component of personal experience and emotions when we talk about religious experiences. He also mentions how any narrative about such religious experiences reflects the subjectivity of the person, including pre-existing ideas, and thinks the religious experience is linked to internal and private feelings (Schmidt 2016, p. 99). His contemporary, Rudolf Otto, defends that a religious experience is a *mysterium tremendum et fascinans*, a feeling of awe and fear in the presence of the wholly Other unique experience, different than other forms of experiences and feelings. As Schmidt mentions, he has been regarded as biased and judgmental. Still, Otto's perspective allows for an understanding of the individual experience and also takes into consideration the set of beliefs and values that provide the cultural framework in which the experience is embedded (Schmidt 2016, p. 101).

We saw how new contemporary spiritualities within a wider setting of "re-enchantment" (Isnart and Testa 2020) are well received in a country where the majority is historically Catholic. Such practices are individualized (Panagiotopoulos and Roussou 2022) but also communitarian (Rocha 2017). They include a concern with one's well-being (both mentally and bodily) but also an uneasiness with the well-being of the planet. All such problems are alleviated through building a strong reconnection with nature in a special place, i.e., Sintra. As a special place, Sintra evokes stories of the glorification of a mythical (even nationalistic) Portuguese past at the same time that it connects people and energies linking to Tibet and India.

As we saw above, the Sintra space elicits such awesome feelings and evokes bursts of spirituality that are interpreted as a link to old traditions of the veneration of Mother Earth, such as re-connecting with nature and listening to one's inner-self. Sintra "quivers with affective energy", to use Thrift's poetical definition (Knott 2010, p. 32). Ammerman stated (Ammerman 2010, p. 158) that "if spirituality of mystery is about the cognitive domain, a spirituality of awe is about affect". Such effects are also what Meyer entitles "sensational forms", binding people together. All such feelings and perspectives arise from what we can name "spiritual elasticity". As Ammerman also explains, the study of religion (and I would add, spirituality) cannot be neatly contained in binary categories of organized vs. individual, religious vs. spiritual, or theistic and transcendent vs. non-theistic and immanent: "(. . .) understanding religion requires that we take spiritualties as seriously as we have always taken belief and belonging" (Ammerman 2013, pp. 33–34). Combining such concepts with this special issue's proposal to think of "transreligosity" (Panagiotopoulos and Roussou 2022) as a conceptual condensation of transgressions of borders between religion and spirituality, religiosity and non-religiosity, and religiosity and well-being or healing leads us to what happens in Sintra. If all these aspects are contained within the discourses about spirituality, Sintra is a good case to apply such intention, since it is a sacred place that attracts all this and therefore an excellent case of spiritual plasticity. Sintra, as a spiritual place, combines a transversality of symbolic spaces of belief, which are extremely porous and hybrid (going from secret tunnels linking Sintra to Tibet or Moorish enchanted princesses to meditation by the sea), transreligious, and "instantiations of transreligiosity" (Panagiotopoulos and Roussou 2022).

The initial quote, taken from the film *Artificial Intelligence*, of the robots affirming the human need to search for "who made them", is, after all, always present, both needing and allowing for a lived spiritual plasticity in the times of crises in which we live.

**Funding:** This research was partially funded by HERA-Humanities in the European Research Area, CRP 5087-00505A, Project HERILIGION-The heritagization of religion and the sacralization of heritage in contemporary Europe grant number [grant number HERA.15.033 and by CY Initiative d'Excellence "Investissements d'Avenir" ANR-16-IDEX-0008, EXPER Project- Experiences of glocalized heritage: imaginaries, appropriations and use conflicts of World heritage historical and religious site.

**Institutional Review Board Statement:** Not applicable.

**Informed Consent Statement:** Informed consent was obtained from all subjects involved in the study.

**Data Availability Statement:** Data from this research is available through publications and also at the HERILIGION site http://heriligion.eu, acessed on 23 June 2023.

**Conflicts of Interest:** The author declares no conflict of interest.

## Notes

1.  The issue of how processes of heritagization interact with various conceptualizations of sacred and secular spaces were discussed by Niedźwiedź and Baraniecka-Olszewska (2020). As I have discussed such implications elsewhere (Saraiva and De Luca 2021; Cardeira da Silva and Saraiva 2022), I will not develop these themes here.
2.  As shown in this text, we have long overcome the Durkheimian strict division between the sacred and the profane. Still, his notion that shared experiences have a special force and that sacred forms both bind individuals together and contribute to group cohesion holds true in Sintra's case, as explained in some of the interview excerpts.
3.  I chose these groups for two main reasons: other materials have been discussed elsewhere (Saraiva 2013; Saraiva and De Luca 2021) and because these groups are the ones whose discourse expands widely on their emic notions of the relation between spirituality and the natural world of Sintra.
4.  Traditional theories hold that the Celtici were a pre-Roman group that included several populi that populated several areas of what is now Portugal. The reference to Celtic traditions related to Mother Nature is nowadays re-invented and valorized by several New Age groups in Portugal.
5.  *Romaria* is a feast in honor of the patron saint.
6.  For a detailed discussion of the archeological studies see Cardim Ribeiro (1998a, 1998b, 2019) and Gonçalves and Santos (2020). Such research has been carried out under the supervision of the Archeological Museum of Odrinhas. See also http://museuarqueologicodeodrinhas.cm-sintra.pt/escavacoes/1/alto-da-vigia.html (accessed on 20 June 2022).
7.  *The Journal of William Beckford: Portugal and Spain* 1787–1788, Alexander Boyd, Portuguese edition with translation and preface by João Gaspar Simões. Lisboa, National Library (1988, p. 152).
8.  I mean *new* in the Portuguese religious context, as explained later.
9.  As several of the interviewees described.
10. This is especially important in the case of Sintra, since it was classified by UNESCO in the category of cultural landscape, encompassing both built and natural heritage. For the cases presented in this text, it is the natural component that is at stake.
11. In luxurious manor houses and palaces, directed at wealthy enterprises that offer such treats to their collaborators.
12. Pseudonyms are used at all times.
13. Sintra is an immense municipality, with an area of almost 320 square meters and 385,606 inhabitants (2021 census) and one of the most culturally and religiously diverse municipalities in the country, where many immigrants from the former colonies and elsewhere live in cities that are dormitories for the workers in the Lisbon area. The paths and sites that spiritual groups frequent are only the core of the UNESCO classified Sintra. Other than foreigners, individuals participating in such activities come from various backgrounds and social classes, with a majority belonging to the middle class and higher social strata.
14. Originally, also health impacts of an open cast mine.
15. In the same vein, Steil and Toniol (2011) approach the experience of hikers on trails as places for restoring forces and energetic fluids for the health of body and soul and thus also show how dichotomies such as mind and body, nature and culture, and subject and object collapse when one reflects upon the therapeutic character of nature walks and their relation to spirituality.
16. Peirano (2003) highlights the place of the strength of religion as a privileged locus of everyday life, which is, for her, an expansion of the senses. This notion dialogues directly with Sointu's and Woodhead's conceptions of the relations between spirituality and well-being, as well as Meyer's notion of sensational forms, discussed in this text.

17      Ley lines are alignments drawn between various historic structures and prominent landmarks. The idea was developed in early 20th-century Europe, with believers arguing that these alignments were recognized by ancient societies that deliberately erected structures along them. Since the 1960s, members of several esoteric traditions have commonly believed that such ley lines demarcate "earth energies" and serve as guides for alien spacecraft.

18      http://www.obod.com.pt/ordem.htm (accessed on 28 June 2023), my translation.

19      Agharta is a legendary kingdom that is said to be located on the inner surface of the Earth, related to the belief in a hollow Earth.

20      My translation.

21      The case of Sintra dialogues with the geomythology discussed by Fotiou (this volume) in Greece, in the way that followers of religious revitalization stress the idea of individuals who engage with universal ideals and worldviews.

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
