# Peer review of "A Sacred Place, an Enchanted Space: Crisis and Spiritual Elasticity in the Mountain of the Moon"

_religions, doi:10.3390/rel14091153_

Round 1

Reviewer 1 Report

This article achieves what it aims to portray, namely “the notion of the ‘magic space’ and the way a space attracts groups and individuals...” I was unfamiliar with this type of research, but this paper is very well documented through extensive research and interviews. The concept of Solastalgia expresses well the feeling of homelessness when one is divorced from natural “places.” This concept gives a universal dimension to these local experiences. For the users as well as this reviewer, “Sintra perfectly embodies the notion of place as... ‘the site of a powerful fusion of self, space and time.’”

I have a problem with the sentence (lines 201-202) “Tweed defends that ‘space’ is a more abstract notion, whereas ‘space’ is a more defined notion, linked directly to lived experience.” Is this the opposition between “religious space” generally, and "local space" linked to experience?

Author Response

Response to Reviewer 1 Comments

thank you very much for your review and comments.

Point 1-  I have a problem with the sentence (lines 201-202) “Tweed defends that ‘space’ is a more abstract notion, whereas ‘space’ is a more defined notion, linked directly to lived experience.” Is this the opposition between “religious space” generally, and "local space" linked to experience?

Response 1- I apologize, and understand your puzzlement. It was a typo. I meant to say "(lines 201-202) “Tweed defends that ‘space’ is a more abstract notion, whereas ‘place’ is a more defined notion, linked directly to lived experience.” That is why I say that Sintra is a ´space` that became a ´place`, and that was built over the centuries and the use of its space as a sacred place. I write this in the following sentence, which comes right after the reference to Tweed: 

"If religious spaces are particular spaces, Sintra is a (religious) space that became a very special place, defined by the multiple experiences of people throughout the centuries".

I have corrected the typo. Hope it is clear now. It is exactly as you suggested in your comment, that is, “religious space” generally, and "local space" linked to experience.

Reviewer 2 Report

This is an excellent paper about a very interesting topic. I believe the paper could be improved with minor revisions even as it is already a valuable contribution.  I have three suggestions:

Firstly, The concept of 'nature' is here explored in all of its polysemic complexity. This might be explicitly clarified.

Secondly, Mircea Eliade's concept of 'numinous spaces' (see The Sacred and the Profane) expresses in many ways the notion that the paper is trying to convey of Sintra as a space which affords experiences of transcendence.  It might be a useful reference here.

Thirdly, at times it is uncertain if 'sacred' is taken as an ethnographic concept or as an analytical tool.  This should be addressed in the light of the criticism that Durkheim's polarity of sacred/profane has received over the decades.

The English is good apart from some minor corrections, mostly typos.

Author Response

Response to Reviewer 2 Comments

This is an excellent paper about a very interesting topic. I believe the paper could be improved with minor revisions even as it is already a valuable contribution.  I have three suggestions:

Thank you for your review and comments. Here are my responses: 

Point 1: Firstly, The concept of 'nature' is here explored in all of its polysemic complexity. This might be explicitly clarified.

Response 1: "Nature" here refers directly to the natural world, in this case,  the natural world of Sintra. This is important also because Sintra has been classified by UNESCO in the category of "Cultural landscape", where the natural (and not only built heritage) is very important. I have attached a paragraph and a footnote explaining this:

As the above examples show, Sintra is a sacred place, generated through layers of spiritual uses and religious occupations; simultaneously, it is also a generative place: Sintra´s space does things (Tweed 2011:229), as we will see. It acts as a “powerful magnet”[i],  attracting a multitude of spiritual groups, individuals who look for inner-knowledge and re-connection with nature to alleviate personal or collective (planetarian) crisis. The concept of ´nature´is here taken at face-value, in the broadest sense, as the physical world or universe. This natural world is not a passive reality, but a natural environment with its own life. It can “do things” to people. This is the case of Sintra, where its natural environment can help and heal people, as we shall see[ii].

[ii] This is specially important in the case of Sintra, since it was classified by UNESCO in the category of Cultural Landscape, encompassing both built and natural heritage. For the cases presented in this text, it is the natural component that is at stake.

Secondly, Mircea Eliade's concept of 'numinous spaces' (see The Sacred and the Profane) expresses in many ways the notion that the paper is trying to convey of Sintra as a space which affords experiences of transcendence.  It might be a useful reference here.

Response 2: done, included!

The sui generis nature of the sacred, relates directly to the experiences, aesthetics and senses. Indeed,  Sintra is a `numinous space' (Eliade 1959) , which conveys experiences of transcendence.  This can be expanded to think of Sintra as a multivocal and polysemic scenario that is both generated and generative (Tweed 2011, p. 121).  Sintra is a space that becomes a special place (Saraiva and De Luca 2021, p.153); this was constructed over time and is reinforced today in both visitors and religious users of the sacred mountain. 

Point 3: Thirdly, at times it is uncertain if 'sacred' is taken as an ethnographic concept or as an analytical tool.  This should be addressed in the light of the criticism that Durkheim's polarity of sacred/profane has received over the decades.

Response 3: sacred is used here mostly as an ethnographic concept, that is, the way people think of and talk about what is "sacred" about Sintra.I do not  intend to discuss Durkheim here. The notion that the secular/profane divide is long overcome is clear in several parts of the text, also using Ammerman´s and woodhead´s references.   I have included some text explaining this, and a footnote on Durkheim:

As shown in this text, we have long overcome the Durkheimian strict division between the sacred and the profane. Still, his notion that shared experiences have a special force and that sacred forms both bind individuals together and contribute to group cohesion holds true in Sintra´s case, as explained in some of the interview excerpts.

Reviewer 3 Report

The article is entitled to be published because it has relevance on the frontiers of religious elements and a “new geography of religion”, based on the city of Cintra. However, I miss features, which I keep listing below.

The first is found in the lack of temporal indication in the introduction of the article, which could organize it better;

The second, when it comes to the issue of spirituality, religion, and everyday life, indicates the entrance of the categories of spirituality and everyday life. However, Peirano highlights the place of the strength of religion as a privileged locus of everyday life. It is for her an expansion of the senses. I believe it is worth mentioning this intense debate in the footnote.

Third, the introduction does not indicate the reason for choosing “the material relating to druids, shamans, Yoga 81 retreats and Eubiose”. Not even the temporality of the “field” and the analyzed groups’ conditions are indicated.

Fourth, with the occurrence of the term “pagan”, it is necessary to explain what is considered within the term, because it can be inclusive of a Christianization of the rituals to consider them as such.

Fifth, likewise, what are the traditions known as Celts? This should be indicated in a footnote. Because the term has a history throughout Europe.

Sixth, what is indicated as “romantic” must be defined as “romantic time”.

Seventh, what are the “Afro-Brazilian” practices? What is being considered if you are taking off this as if you are writing? For such indications on whitening and its cultural migrations, see the text by R. Prandi, Social References of Afro-Brazilian Religions: Syncretism, Whitening, Africanization, Horizontes Antrpologicos 4 (8), 1998: https://www.scielo.br/j/ha/a/g35m5TSrGjDp9HxYGjBqNGg/#.

The eighth point to be considered: terms like “the woods are attesting to the Glory of the Creator” should be rewritten for some care with the language of devotion genre. Parts like these should be completely rewritten;

The ninth fragment: “Although the Portuguese context is quite different from the US one, such statements are still valid, and I fully agree that religious pluralism is “endemic in the human condition”, special care must be taken with the statement. In the same way, the eighth point indicates a connection with the religious expressions of Cintra. There is a tendency towards religious plurality when the hegemonic traditions do not have such an internal emblem. On an internal level, the will to influence different areas of life is valid. There is also a datum of the author's identification with the Cintra phenomenon and a lack of a social class focus in the analysis. The writing is very close to the experiences found in Cintra, without showing the contradictions and internal disputes. This should be further highlighted at least in the footnote of the article. In this case, does the question whether people who need to survive in the face of violence in European cities have the same relationship as those who frequent Cintra? This element is invisible in the article.

The tenth point to be drawn touches on the issue of “spirituality”: in the fragment “This is what, for me, is spirituality”, when it reflects on spiritualities in a different way. There is a set of new spiritualities being reflected on the rural parameter, on access to nature. For example, the material relates to a spirituality of pilgrimages in which R. Toniol and C. Steill, write in their “Ecology, Body and Spirituality: an Ethnography of Ecological Walking Experiences in a group of eco-tourists”, Cadernos CRH, v. 24, 2011: https://www.scielo.br/j/ccrh/a/SRVtB8PWLN8pkdNP95WMtpn/abstract/?lang=pt. Another form of ruralization of religious experiences was described by F. Py and M. Pedlowski in the article “Pentecostalization settled in the Settlement Zumbi dos Palmares”, Perspectiva Teológica, v. 52: https://www.scielo.br/j/pteo/a/LMch3RcxnWhFyJYDRxW5BzM/?lang=pt. Indeed, Pentecostal associations have been influencing a new path of insertion in the agrarian environment. Both materials are concerned with describing forms of “spiritual” experience in rural areas and provide an overview of spirituality. These materials are important to be attached to the article in a footnote and indicate that there are different types of approaches to spiritualities in the rural environment that are not present in the work.

Thus, the article deserves to be published, but before that, it needs to be rewritten in the marked parts, in the attachment of the footnotes next to the ten points marked here.

Author Response

Thank you for your review and helpful comments, as well as the suggestions for adding some footnotes and references. Here are my responses:

Point 1: The first is found in the lack of temporal indication in the introduction of the article, which could organize it better;

Response 1: The 3rd to last paragraph in the introduction clearly states the temporal scope of the article, explaining that it is based on fieldwork carried on from 2017 to the present, but also using archival and historical material on Sintar and on the temporal layers that have made Sintra what it is today.  I have added the historical material:

This paper is based on archival research, historical documents and fieldwork in Sintra, carried on from 2017 to the present date, which comprised participant observation in walks and tours with the various religious/spiritual groups, and in depth interviews with their leaders, guides, and followers. 

Point 2: The second, when it comes to the issue of spirituality, religion, and everyday life, indicates the entrance of the categories of spirituality and everyday life. However, Peirano highlights the place of the strength of religion as a privileged locus of everyday life. It is for her an expansion of the senses. I believe it is worth mentioning this intense debate in the footnote.

Response 2: yes, you are right. Done, here is the footnote: 

Peirano (2003) highlights the place of the strength of religion as a privileged locus of everyday life, which is, for her,  an expansion of the senses. This notion dialogues directly with Sointu´s and Woodhead´s conceptions of the relations between spirituality and well-being, as well as Meyer´s notion of sensational forms, discussed in this text.

Point 3: Third, the introduction does not indicate the reason for choosing “the material relating to druids, shamans, Yoga 81 retreats and Eubiose”. Not even the temporality of the “field” and the analyzed groups’ conditions are indicated.

Response 3: The temporality of the filed is indicated in the above mentioned paragraph of the introduction. The reason for choosing thse groups has been added as a footnote: 

I chose these groups for two main reasons: other materials have been discusses elsewhere (Saraiva 2023; Saraiva and De Luca 2021), and because these groups are the ones whose discourse expands widely on their emic notions of the relation between spirituality and the natural world of Sintra.

Point 4. Fourth, with the occurrence of the term “pagan”, it is necessary to explain what is considered within the term, because it can be inclusive of a Christianization of the rituals to consider them as such.

Response 4: exactly, you are totally right, and the ration between former pagan sites and their Christianization is explained in section 2., also using references (as Blanes and Mapril 2013)  that discuss how this happened in Southern europe.

Point 5. Fifth, likewise, what are the traditions known as Celts? This should be indicated in a footnote. Because the term has a history throughout Europe.

Response 5. Right, I have added a footnote and  this is also explained in the reference cited Cardim Ribeiro, which expands on the Celts in Portugal.  I had forgotten to insert it also when I refer to the Celts concerning Santa Eufemia. I have now included it. Here is the footnote:

Traditional theories hold that the Celtici were a pre-roman group that included several populi, which populated several areas of what is now Portugal.  The reference to Celtic traditions related to mother nature is nowadays re-invented and valorized by several New Age groups in Portugal.

Point 6. Sixth, what is indicated as “romantic” must be defined as “romantic time”.

Response 6: right, this is done. Where I mean the time period, I always use "romatic period" or "romantic era". In other places I use Romanticism as a literary and cultural movement,

Point 7: Seventh, what are the “Afro-Brazilian” practices? What is being considered if you are taking off this as if you are writing? For such indications on whitening and its cultural migrations, see the text by R. Prandi, Social References of Afro-Brazilian Religions: Syncretism, Whitening, Africanization, Horizontes Antrpologicos 4 (8), 1998: https://www.scielo.br/j/ha/a/g35m5TSrGjDp9HxYGjBqNGg/#.

Response 7: I have extensively worked on the expansion of the Afro-Brazilian religions in Portugal (see Saraiva 2008, 2010, 2011,  2013, 2016, 2020). I have also written about the Afro-Brazilian religions in the Sintra area (Sraaiva 2023, 2020).  I know Prandi´s work (and other authors) well. I have not expanded on these here as this is not within the scope of this text. 

Point 8: The eighth point to be considered: terms like “the woods are attesting to the Glory of the Creator” should be rewritten for some care with the language of devotion genre. Parts like these should be completely rewritten;

Response 8: this sentence was written specifically as a metaphor, when comparing to kitsch images of calendars, and it should not be taken at face value.  That is why I write:

"One enters a magical and wonderful route of connection with nature that also resembles the vaguely kitsch photos in a religious calendar or book: a few rays of sun coming through the woods and attesting to the Glory of the Creator".

Point 9: The ninth fragment: “Although the Portuguese context is quite different from the US one, such statements are still valid, and I fully agree that religious pluralism is “endemic in the human condition”, special care must be taken with the statement. In the same way, the eighth point indicates a connection with the religious expressions of Cintra. There is a tendency towards religious plurality when the hegemonic traditions do not have such an internal emblem. On an internal level, the will to influence different areas of life is valid. There is also a datum of the author's identification with the Cintra phenomenon and a lack of a social class focus in the analysis. The writing is very close to the experiences found in Cintra, without showing the contradictions and internal disputes. This should be further highlighted at least in the footnote of the article. In this case, does the question whether people who need to survive in the face of violence in European cities have the same relationship as those who frequent Cintra? This element is invisible in the article.

Point 9: thank you for all this; I have added a footnote addressing the issues you address; , but, for instance, questions of violence in cities is beyond the scope of the article. 

Footnote:  Sintra is an immense municipality, with an area of almost 320 square meters and   385 606 inhabitants (2021 census), and one of the most culturally and religiously diverse in the country, where many immigrants from the former colonies and elsewhere live, in cities that are dormitories for the workers in the Lisbon area. The paths and sites that spiritual groups frequent is only the core of the UNESCO classified Sintra. Besides foreigners, individuals participating in such activities come from various backgrounds and social classes but with a majority belonging to middle class and higher social strata.

Point 10: The tenth point to be drawn touches on the issue of “spirituality”: in the fragment “This is what, for me, is spirituality”, when it reflects on spiritualities in a different way. There is a set of new spiritualities being reflected on the rural parameter, on access to nature. For example, the material relates to a spirituality of pilgrimages in which R. Toniol and C. Steill, write in their “Ecology, Body and Spirituality: an Ethnography of Ecological Walking Experiences in a group of eco-tourists”, Cadernos CRH, v. 24, 2011: https://www.scielo.br/j/ccrh/a/SRVtB8PWLN8pkdNP95WMtpn/abstract/?lang=pt. Another form of ruralization of religious experiences was described by F. Py and M. Pedlowski in the article “Pentecostalization settled in the Settlement Zumbi dos Palmares”, Perspectiva Teológica, v. 52: https://www.scielo.br/j/pteo/a/LMch3RcxnWhFyJYDRxW5BzM/?lang=pt. Indeed, Pentecostal associations have been influencing a new path of insertion in the agrarian environment. Both materials are concerned with describing forms of “spiritual” experience in rural areas and provide an overview of spirituality. These materials are important to be attached to the article in a footnote and indicate that there are different types of approaches to spiritualities in the rural environment that are not present in the work.

Response 10. Thank you for the alert. Following your comments, I have read, commented on   Steil and Toniol useful and interesting text and  added it to the references. The issue of spiritualities in the rural areas does not apply to Sintra, which is NOT AT ALL a rural area.

footnote: 

In the same vein, Steil and Toniol (2011) approach the experience of hikers on trails , as places for restoring forces, energetic fluids for the health of body and soul, and thus also show how dichotomies such as mind and body, nature and culture, subject and object collapse when one reflects upon the therapeutic character of nature walks and its relation to spirituality.

Round 2

Reviewer 3 Report

To provide a new dimension of the rural aspect of the paper, the description below is important, which was not considered by the authors:

"Another form of ruralization of religious experiences was described by F. Py and M. Pedlowski in the article “Pentecostalization settled in the Settlement Zumbi dos Palmares”, Perspectiva Teológica, v. 52: https://www.scielo.br/j/pteo/a/LMch3RcxnWhFyJYDRxW5BzM/?lang=pt. Indeed, Pentecostal associations have been influencing a new path of insertion in the agrarian environment. Both materials are concerned with describing forms of “spiritual” experience in rural areas and provide an overview of spirituality. These materials are important to be attached to the article in a footnote and indicate that there are different types of approaches to spiritualities in the rural environment that are not present in the work"

Author Response

Point 1:

To provide a new dimension of the rural aspect of the paper, the description below is important, which was not considered by the authors:

"Another form of ruralization of religious experiences was described by F. Py and M. Pedlowski in the article “Pentecostalization settled in the Settlement Zumbi dos Palmares”, Perspectiva Teológica, v. 52: https://www.scielo.br/j/pteo/a/LMch3RcxnWhFyJYDRxW5BzM/?lang=pt. Indeed, Pentecostal associations have been influencing a new path of insertion in the agrarian environment. Both materials are concerned with describing forms of “spiritual” experience in rural areas and provide an overview of spirituality. These materials are important to be attached to the article in a footnote and indicate that there are different types of approaches to spiritualities in the rural environment that are not present in the work"

Response 1: the reason I did not mention the referred text is because it has nothing to do with my case study. Sintra is not a rural milieu at all, and there are no issues of community religious mobility in the cases presented in my text. I know the literature on religious mobility in Brazil, but it does not relate to this case study.